# COVID-19 and Inherited Metabolic Disorders: One-Year Experience of a Referral Center

**DOI:** 10.3390/children8090781

**Published:** 2021-09-06

**Authors:** Albina Tummolo, Giulia Paterno, Annamaria Dicintio, Pasquale Stefanizzi, Livio Melpignano, Maurizio Aricò

**Affiliations:** 1Department of Metabolic Diseases and Clinical Genetics, Giovanni XXIII Children Hospital, Azienda Ospedaliero-Universitaria Consorziale Policlinico, 70126 Bari, Italy; giupatvi@gmail.com (G.P.); annamaria.dicintio@gmail.com (A.D.); 2Department of Biomedical Science and Human Oncology, Post Graduate School of Hygiene and Preventive Medicine, University of Bari Aldo Moro, 70124 Bari, Italy; pasquale.stefanizzi@uniba.it; 3Medical Direction, Giovanni XXIII Children Hospital, Azienda Ospedaliero-Universitaria Consorziale Policlinico, 70126 Bari, Italy; livio.melpignano@policlinico.ba.it; 4Strategic Control Azienda Ospedaliero-Universitaria Consorziale Policlinico, Piazza Giulio Cesare, 70124 Bari, Italy; maurizio.arico@policlinico.ba.it

**Keywords:** COVID-19, inherited metabolic disorders, rare diseases, SARS-CoV-2

## Abstract

Understanding the potential risks of patients with inherited metabolic disorder (IMD) exposed to the COVID-19 pandemic is an unmet need for those involved in their management. Here, we report on the incidence of COVID-19 in a cohort of patients with IMD treated at a children’s hospital and compare them with a matched control group. Among the total number of 272 patients actively followed at a referral center, 19 (7%) tested positive for SARS-CoV-2 between March 2020 and March 2021. Their median age was 16.2 years (range 1.4–32.8 years). In two-thirds of the cases, the source of infection was a family member; 12/19 patients (63%) were asymptomatic, only one required hospitalization, and none of them died. In our single-center experience, COVID-19 had a moderate impact on a relatively large cohort of patients with IMD, including children and young adults. The clinical course was very mild in all but one case. The proportion of symptomatic cases and the clinical course were comparable in patients with IMD and in a group of matched, non-IMD COVID-19 controls from the general population.

## 1. Introduction

Rare diseases are diseases that affect a small number of people compared to the general population. Despite their rarity, these conditions collectively affect ~30 million people across Europe [1] and come with particular challenges related to a higher risk of serious illness like COVID-19-related disease. Small numbers and limited experience may hamper the confidence of the attending specialist when facing this situation.

Inherited metabolic disorders (IMD) represent one of the most common groups of Mendelian diseases [2]. Indeed, despite being individually rare or ultra-rare (<1:2,000,000), they are increasingly recognized all over the world thanks to neonatal screening programs and improved diagnostic methods [3].

In the course of events such as the COVID-19 pandemic, patients with IMD may represent a group of fragile subjects with an increased risk or at least special needs.

In Italy, the spread of the pandemic occurred earlier than in other European countries. Since then, the frequency of the virus variants has changed over time, with the alpha variant of the virus [4] becoming progressively prevalent in the country, reaching an 88.1% prevalence by March 2021 [5].

The available reports on the impact of the pandemic on the management of IMD patients [6,7] have emphasized the effects on clinical planning, access to therapies, and management of emergencies. Few studies have so far focused on the clinical impact of SARS-CoV-2 infection on this group of patients.

Here, we report on the incidence of COVID-19 in a cohort of children and young adult patients with IMDs cared for at a children’s hospital. We aim for characterizing the frequency, the course, and the severity of COVID-19; its impact on the course of the underlying disease; and its management.

## 2. Materials and Methods

All patients with IMD followed at our unit who tested positive for SARS-CoV-2 infection between March 2020 and March 2021 were included. No local ethics committee approval was needed, as this was an observational study. The diagnosis was based on the positivity of the nasopharyngeal swab analyzed by real-time reverse-transcription polymerase chain reaction (rRT-PCR). Swabs were performed either under the direction of the general practitioner, as part of the tracing of social/familial contacts, or as part of the screening program upon patient admission. A communication of positivity to external testing was given by email. Information on the clinical status and follow-up was collected by telephone interview or teleconsulting. During the national lockdown period, biochemical monitoring was performed at home, the results were reported to the clinical center, and the treatment was modified accordingly.

Then, to evaluate the possible differences compared to the general population, we retrieved data on the subjects who tested positive for COVID19 in the same time interval from the health surveillance database of the Apulian region. Subjects with COVID-19 were matched with IMD patients by propensity score matching. The variables used to compute the propensity scores were age and sex.

By the institutional policy, given the retrospective and noninterventional design of the study, formal approval by our institutional review board was not considered necessary.

## 3. Results

Among a total of 272 patients with IMD treated and followed-up on over the last two years at our referral center, 19 patients (7%): 12 females and 7 males, with a median age of 16.2 years (range 1.4–32.8 years), tested positive for SARS-CoV-2. In 12 out of the 19 cases (63%), the source of infection was a family member (Table 1). All the cases occurred between October 2020 and March 2021. All of them were documented as infected by the alpha variant of SARS-CoV-2. None of them were vaccinated against SARS-CoV-2.

In 5/19 cases, the scheduled visit at the referral center was postponed because of COVID-19. Two patients were affected by lysosomal storage disorders (LSD) and nine patients by altered phenylalanine metabolism: three by hyperphenylalaninemia (PHA) and six by phenylketonuria (PKU). Five patients were affected by urea cycle disorders (UCD), and the other three patients had cobalamin C deficiency, type 2 glutaric aciduria, and maple syrup urine disease (MSUD) (Table 1).

Of the two patients with lysosomal storage disease—namely, Gaucher type 1 and juvenile Pompe disease, the first one developed only rhinitis, and the enzyme replacement therapy (ERT) was not modified. The second one, with long-lasting restrictive respiratory syndrome requiring nocturnal noninvasive ventilation, developed a fever as the only COVID-related manifestation and could be treated at home. The ERT was modified with the omission of one scheduled injection.

Among the nine patients (47%) affected by altered phenylalanine metabolism, none of them showed any increase in Phe plasma values; therefore, a reduction in Phe intake was not necessary in those with protein-restricted diets or in the case of a fever.

Among the other patients with disorders of intermediate metabolisms with decompensation risks, only the patient with MSUD (OMIM # 248600), aged 26.8 years, experienced acute metabolic decompensation. She presented with a fever (>38 °C) and inappetence, which resulted in a progressive reduction of her daily protein intake. Ten days after the diagnosis of COVID-19, she developed altered consciousness, hallucinations, and vomiting; her leucine levels peaked at 708 µmol/L, confirming an acute metabolic imbalance. Hospitalization was required for seven days, with an enteral emergency diet intravenous infusion of fluids, as well as nonprotein calories.

The comparison of this group of 19 patients with IMD with 34 COVID-19-positive control subjects matched by propensity scores revealed that the proportion of asymptomatic subjects was 76% in the control group vs. 63% in IMD patients (*p*-value = 0.32); no hospitalizations occurred in the control group (*p*-value = 0.97); no deaths were reported in both groups (Table 2).

## 4. Discussion

Understanding the special needs and potential risks of patients with IMD exposed to the COVID-19 pandemic has been an unmet need for the pediatricians involved in their treatment. Thus, we revised our experience on patients followed-up over the last two years at our referral center. Of them, only 19, i.e., 7%, developed COVID-19.

The source of infection was familial/social exposure in two-thirds of them. This finding is widely in keeping with the trend observed in our hospital for children with COVID-19 [8] but, also, for healthcare workers, suggesting that the application of a stringent policy for DPI use and distancing turned out to be effective in preventing the intrahospital spread of SARS-CoV-2 infection. [9]

The manifestation of SARS-CoV-2 infection is much less severe in children and adolescents than in adults and the elderly population [10]. In this series of patients with IMD, the manifestation of COVID-19 was very mild, with only one case requiring hospital admission, and none of them requiring admission to the ICU. This may be explained by the young ages of our population, in which the clinical manifestations mirrored that of the general population of the same age group. Furthermore, the fact that all the cases were concentrated in the second and third pandemic waves of our country is in line with the general trend of the incidences in the pediatric and young adult individuals in the Apulian population [8], which has seen an increase in the index of contagion and hospitalization starting from October 2020 [11].

Two patients with lysosomal storage disease on ERT contracted SARS-CoV-2 infection in a social environment and had a very favorable clinical course. In a recent study, 88 out of 181 patients with Gaucher disease were tested for SARS-CoV-2; of them, 16 (18%) tested positive. None of them required COVID-19-directed treatment or died [12].

Our juvenile-onset Pompe patient, with pre-existent respiratory impairment and nocturnal ventilation support, did not show any further respiratory deterioration. The clinical picture led to a very short ERT discontinuation, in keeping with the recommendation of Wenninger et al., who, in a prospective cohort study of 12 patients with late-onset Pompe disease, showed that the interruption of ERT was associated with a deterioration in the core clinical outcome measures [13].

Among the patients with IMD with decompensation risk, MSUD represented the only disorder with decompensation requiring hospitalization because of a reduction in protein intake and fever, well-known causes of decompensation in these patients. Interestingly, the patient with type 2 glutaric aciduria and the one with arginine succinic aciduria aged 17.6 years in 2019 developed, respectively, two and four decompensations due to intercurrent illnesses, requiring hospitalization. Other patients with IMD with decompensation risk, such as mitochondrial diseases, did not develop COVID19 over the study period. The overall number of admissions/referrals to the center due to intercurrent infections diminished compared to the previous year, likely due to the lockdown social restrictions.

In our center, we followed a large cohort of 183 subjects, children and adults, with Phe metabolism disorders, of which nine (4.9%) were diagnosed with COVID-19. It is interesting to note that a fever was not associated with increased plasma Phe values during regular monitoring; thus, an adjustment of the Phenylalanine intake was not required.

Within the small world of children (and adults) with rare diseases, the role of the referral center has been one of mainly supporting the spokes represented by local hospitals—on one hand, confirming the minimal need for treatment adjustment, but in the meanwhile, offering support for the FAQs and worries of the parents and the adolescents facing an unknown event perceived as potentially catastrophic. Confirming that the usual diet and long-lasting therapies were adequate in this condition turned out to be of help and relief in most cases. In keeping with this approach, the European Society of Endocrinology (ESE) Rare Disease Committee, alongside the Endocrinology European Reference Network (ENDO-ERN), have engaged in an initiative to collect essential data concerning specific groups of patients with rare endocrine conditions who develop COVID-19 [14].

This study has some limitations. Since we did not perform a universal screening of all our patients by nasopharyngeal swab, we cannot rule out that other patients may have developed asymptomatic SARS-CoV-2 infections. Thus, the incidences we report of symptomatic COVID-19 in patients with IMD have to be taken as the minimal amount. Second, not all the patients were directly cared for in our hospital during the manifestation of COVID-19; yet, these were so few that a reporting bias should not represent a major issue. A further study on the effects of other variants of COVID-19 on these patient populations is recommended.

In conclusion, our experience suggests that the COVID-19 circulation among young patients with IMD was minimal, followed by the epidemiologic distribution of the infection and the severity of the manifestations in the general population of the same age. The accurate monitoring and close interactions between the referral centers, the local spokes and patient charities may be instrumental in reassuring the patients and improving the outcomes.

## Figures and Tables

**Table 1 children-08-00781-t001:** Patients and infection characteristics.

Case	Age (y)/Gender	Disease	Traced Contact	Time of Infection	COVID Manifestations; Duration	Impact on Therapy/Diet
1	2.8/F	Hyperphenylalaninemia	Family	October 2020	1-day fever and diarrhea; 6-day diffused eritemato-papular rash	free diet
2	2.5/F	Hyperphenylalaninemia	Family	November 2020	None	free diet
3	15.3/M	Ornitin Transcarbamilase deficiency	Family	November 2020	none	no changes
4	16.5/F	Lysinuric protein intolerance	Family	December 2020	None	no changes
5	12/M	Type 2 glutaric aciduria	Family	December 2020	None	no change
6	4.6/F	Hyperphenylalaninemia	Family	December 2020	None	free diet
7	20.3/F	Cobalamin C deficiency	Family	December 2020	1-day fever, cough, and rhinitis	no changes
8	20.4/M	Phenylketonuria	Work	December 2020	1-day fever and headache	no changes
9	11.9/F	Phenylketonuria	Family	December 2020	none	no changes
10	7.8/F	Phenylketonuria	Family	December 2020	none	no changes
11	24.3/M	Juvenile Pompe disease	Social	December 2020	3-day fever	1 ERT infusion missed
12	29.9/M	Phenylketonuria	Work	December 2020	1-day osteoarticular pain, 7 days loss of taste	no changes
13	17.6/M	Arginine succinic aciduria	Family	January 2021	none	no changes
14	32.8/F	Gaucher Disease type 1	work	February 2021	3-day rhinitis	no ERT infusion missed
15	1.4/M	Arginine succinic aciduria	Family	February 2021	none	no changes
16	26.8/F	Maple Syrup Urine Disease	Family	February 2021	2-day fever, lack of appetite	emergency diet regimen
17	28.1/F	Phenylketonuria	Family	February 2021	none	no changes
18	1.6/F	Citrullinemia type 1	Family	March 2021	none	no changes
19	16.2/F	Phenylketonuria	Family	March 2021	none	no changes

**Table 2 children-08-00781-t002:** Comparison of the main parameters between the IMD and control groups.

	IMD/COVID+	Control Group COVID+	*p*-Value
Number	19	34	−
Sex female, number (%)	12 (63%)	22 (64%)	0.92
Age, median (range)	16.2 (1.4–32.8)	16 (1.2–33.5)	0.32
Asymptomatic, number (%)	12 (63%)	26 (76%)	0.32
Hospitalization, number (%)	1 (5%)	0 (0%)	0.97
Deaths, number (%)	0 (0%)	0 (0%)	−

## Data Availability

The data presented in this study are available on request from the corresponding author.

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
