# Peer review of "COVID-19 and Inherited Metabolic Disorders: One-Year Experience of a Referral Center"

_children, 2021, doi:10.3390/children8090781_

Round 1

Reviewer 1 Report

This is an important paper that summarizes the experience of a medium size metabolic center with respect to metabolic decompensation and other effects of infection with SARS-CoV2 on their patient population during the first year of the pandemic. Interestingly, and in line with anecdotal information for pts with IEM, patients overall did well, with only one decompensation in a patient with MSUD due to the viral infection and symptoms from this. In contrast multiple patients with urea cycle disorders, also are risk of decompensation did not have decompensation.

I have the following questions:

Does the center follow any patients with mitochondrial disease, eg Leigh, MELAS, MERRF, NARP, etc, this is an important IEM population with possible decompensation in illness. If so, what were the results for them?

It appears that other infections were reduced due to infection protocols used for COVID, is it possible to compare the number of interventions/decompensations in the year cited to the year prior? And, a related question, were there decompensations due to non-COVID illnesses in the pandemic and did the number of these change compared to the prior year?

Author Response

The center follows mitochondrial disorders, and in our experience they did not decompensate during the study period, nor were infected by SARS-CoV-2. Interestingly, the overall number of admission/referral to our center due to intercurrent infections diminished compared to the previous year, possibly due to lock-down social restrictions. We have specified this aspect in the discussion.

Reviewer 2 Report

This study is an observational study, describing the natural course of metabolic disorder patients who were infected by COVID-19 during the first wave of pandemic.

There are a few important points; 1. the severity, mortality rate as well as demographic distribution of COVID-19 infection (I believe all of patients were infected by the original variant SAR CoV2) in metabolic patients were no different from non metabolic patients.

2. home management was adequate for mild COVID-19 infection cases

Suggestions 1. summarize type of metabolic conditions included in this study, e.g 2 with LSD, 9 with PKU. Line 80, I had to guess what disorders all 9 patients had

2. mention the severity of metabolic condition, number of patients with severe baseline such as number of hospitalization before COVID-19 pandemic.

2. recommendation for future study. comment the impact of delta variant on metabolic patients

Author Response

I confirm that all the patients were infected by the alpha variant of SARS-CoV-2. This has been specified in the text.

The types of conditions included in the study have been summarized in the Results. I have also mentioned the cases of two patients with severe clinical patterns who decompensated very frequently during 2019, due to other intercurrent illnesses. The opportunity of a further study on the effect of other virus variants on this population is also mentioned in the conclusion.

Reviewer 3 Report

In the current manuscript, the authors have described the incidences of COVID-19 in children and young patients with inherited metabolic disorders. Data also indicate that the requirement for hospitalization is minimal in patients with IMD. Also, no death was reported in patients with IMD due to COVID-19.

-This study is based on the data from one hospital. This study can be more attractive and useful if the IMD patient's data has been included from other hospitals too.  Please let us know the possibilities to include the desired data. 

-Data has been acquired between March 20-21. The authors should mention the type of COVID-19 variants at that time which were spreading. Accordingly, the introduction and discussion part of the manuscript has to be modified in the context of their effects on patients with IMD. 

Author Response

We acknowledge that broadening this study to other reference centers could have been of interest. Unfortunately, such a survey is not feasible at this stage.

As required, the SARS-CoV-2 variant has been specified in the introduction and methods sections, and then discussed accordingly.